# Caregiver-Reported Economic Impacts of Pediatric Rare Diseases—A Scoping Review

**DOI:** 10.3390/healthcare12242578

**Published:** 2024-12-21

**Authors:** Niamh Buckle, Orla Doyle, Naonori Kodate, Melissa Kinch, Suja Somanadhan

**Affiliations:** 1School of Nursing, Midwifery and Health Systems, University College Dublin, D04 V1W8 Dublin, Leinster, Ireland; melissa.kinch@ucdconnect.ie (M.K.); suja.somanadhan@ucd.ie (S.S.); 2School of Economics, University College Dublin, D04 N9Y1 Dublin, Leinster, Ireland; orla.doyle@ucd.ie; 3School of Social Policy, Social Work and Social Justice, University College Dublin, D04 N9Y1 Dublin, Leinster, Ireland; naonori.kodate@ucd.ie; 4UCD Centre for Interdisciplinary Research, Education and Innovation in Health Systems (UCD IRIS), University College Dublin, D04 V1W8 Dublin, Leinster, Ireland

**Keywords:** economic impact, healthcare cost, indirect cost, non-healthcare cost, pediatric, rare disease

## Abstract

**Background/Objectives**: Rare diseases are conditions that are individually rare but collectively common. These diseases can incur significant direct and indirect costs with a combination of high medical expenses, loss of income, and additional non-medical costs. Despite this, research into the economic cost for families of children with a rare disease is lacking. This scoping review aimed to document the evidence on the economic impacts of living with a rare disease for children and their families. **Methods**: Six electronic databases were searched to identify relevant peer-reviewed literature that discussed the family costs (direct medical, direct non-medical, and indirect) of having a child with a rare disease, published between January 1983 and April 2023. The geographical location, type of rare disease, and language were not limited. Data were extracted from the included studies following the screening process and are reported following the PAGER framework for reporting scoping review results. **Results**: The final analysis included 28 studies. The studies highlighted areas of high costs, including visits to healthcare professionals (*n* = 36), medication costs (*n* = 11), presenteeism (*n* = 17), and informal care (*n* = 11). However, gaps in the existing research, such as the focus on metabolic or musculoskeletal rare diseases and the lack of a distinction between rare and ultra-rare diseases, were apparent. **Conclusions**: Having a child with a rare disease can significantly impact a family’s financial health, and these costs extend beyond healthcare costs. Understanding the costs experienced by the rare disease population is important to better define and comprehend the economic impact of rare diseases.

## 1. Introduction

Rare diseases are associated with a high social and economic burden for patients, their families and caregivers, healthcare systems, and society [1,2]. A rare disease is one that affects less than 5 people per 10,000 in the European Union [3], and less than 200,000 people in the United States of America [4]. There are further classifications of rare diseases, with ultra-rare diseases affecting fewer than 1 person in 50,000, and other classifications such as hyper-rare for diseases that affect <1/10^8^ people [5]. These illnesses, of which there are over 6000–10,000 [3,5], are often incurable, and require long-term care management and supportive therapies. With 80% of rare diseases having a genetic component, and 95% lacking treatment options [6], interventions for these illnesses are limited and often costly. Orphan drugs explicitly developed for rare diseases can cost significantly more than non-orphan drugs [7], which can contribute to high prices, poor treatment access, and increased health burden and premature death. Other economic impacts of the lack of treatments and increased care management needs include absenteeism (workdays missed) and presenteeism (reduced productivity) among carers, who are often juggling their care duties with employment [4,8]. Furthermore, the lack of standardization in care approaches internationally means that families experience inequities in rare disease care, which may lead to differing economic experiences for families of children with the same rare disease between different contexts.

The economic costs of rare diseases focus on direct healthcare costs, such as hospitalization or specialized care; direct non-healthcare costs, such as education costs; or indirect costs, such as loss of income or productivity [9,10]. Despite the economic impact of rare diseases on families being a heretofore neglected aspect of the literature on rare diseases, the studies that have been undertaken demonstrate that there are higher per-person costs for children than for adults with rare diseases [4,10]. This may be because rare diseases that affect children may be more acute or complicated than those that survive to adulthood. Another explanation may be because adults with rare diseases can avail of more state benefits and thus can offset the family costs in a way children cannot [11]. Research has demonstrated that caring for children with rare diseases can place significant pressures on caregivers, with studies reporting their experiences of emotional problems, lack of support, reduced income [12], difficulty navigating insurance provisions [13], and moderate to severe caregiver burden [14] across different country contexts. Care tasks in addition to parenting tasks, the lengthy diagnostic process and emotional consequences of the lengthy diagnostic process, and quest for treatment can add further complexity for parents of children with rare diseases compared with those caring for adults [15]. Further research is needed on the economic impact of rare childhood diseases for families so these impacts can be documented and addressed in the future.

A scoping review was selected due to the broad nature of the question and the function of scoping reviews of mapping the available literature. A preliminary search of MEDLINE and the Cochrane Library was conducted to identify any existing reviews on a similar topic. While there are reviews on the economic impact of rare diseases in all populations [1,16], none focus solely on childhood rare diseases and the family perspective of having a child with a rare disease. This scoping review addresses this gap by exploring the direct and indirect economic impacts faced by families that have a child with a rare disease.

This scoping review sought to answer the question, “How is the economic impact (direct and indirect costs) for families and caregivers of children with a rare disease identified and measured in the literature?”. Answering this question involved several sub-objectives, which influenced the data charting and reporting of results as follows:Identify, appraise, and synthesize knowledge surrounding the economic impacts of rare diseases affecting children and their families by considering healthcare and non-healthcare direct and indirect costs.Understand how these healthcare and non-healthcare direct and indirect costs are measured (e.g., validated tools, economic costing approaches).Clarify what specific disease populations and disease characteristics are most researched.Determine the study settings, rare conditions and geographical contexts, and the study types and organizations involved (e.g., charitable organizations, pharmaceutical companies, etc.)Highlight any gaps in the literature on the economic impact of rare diseases on children and their families.

## 2. Materials and Methods

This scoping review was conducted following the Arksey and O’Malley [17] methodological framework and guidance from the Joanna Briggs Institute [18]. The protocol for this review was presented to PPI representatives and knowledge users working in the rare disease field for their review and feedback. The peer-reviewed protocol is available to read in the protocol set out by Buckle et al. [19].

### 2.1. Inclusion and Exclusion Criteria

Literature published in English from 1983 to 4 April 2023 was collected from six databases: EconLit (EBSCO), ABI/Inform Global (Proquest), MEDLINE (Ovid), PubMed, CINAHL Plus, and Scopus (Elsevier). A search strategy using index terms such as “family”, “caregiver”, “economic impact”, “cost of illness”, “child”, “pediatric”, “rare disease”, and “orphan disease” was utilized to capture relevant studies. The full search strategy for each database is included in the Appendix A accompanying this article.

No language restrictions were placed on the search strategy, and studies in languages other than English were screened using Google Translate powered by Google Cloud Translation. The publication data limit of 1983–present reflected the increased focus on rare disease research since the introduction of the 1983 Orphan Drug Act in the United States [20]. No limits were placed on geographical location. As mentioned above, previous scoping reviews found little research on the economic impacts of childhood rare diseases on families for inclusion. In recognition of this, studies with adults included in their study population were not immediately excluded but were only retained if the study isolated and presented the results of the economic impact on children and their families separately to those of adults.

Original studies of any design that provided family-, parent-, or caregiver-reported impacts on the economic costs (direct and/or indirect) of pediatric rare diseases and met the inclusion criteria were eligible for inclusion. Secondary research or non-peer-reviewed research, including reviews, conference materials, dissertations/theses, etc., were excluded.

### 2.2. Procedure

The title and abstract screening and full-text screening was performed by two authors (NB and MK), and any conflicts were resolved through discussion with a third author (SS). Data charting was completed by NB and audited by SS according to a data charting template devised by the primary author (NB) and reviewed by the supervising authors (OD, NK, SS). Any discrepancies in data charting were resolved through discussion. Data charting was also supported by NVIVO 20 [21], to confirm the extracted data. This template is available in the associated scoping review protocol [19].

### 2.3. Quality Assessment

As reported in the protocol, a quality appraisal using the Consolidated Health Economic Evaluation Reporting Standards 2022 (CHEERS) [22] was initiated prior to data charting to determine the level of quality of the included studies. This appraisal method focuses on the methodological quality of economic evaluations and was selected in anticipation of sourcing these types of articles during the search process. However, as most included studies were cross-sectional studies with a cost-of-illness focus, the CHEERS tool was not appropriate for quality assessment. A quality assessment of included articles was instead performed using the Mixed Methods Appraisal Tool (MMAT) [23]. As recommended by this tool, no overall score was calculated; the full assessment is included in the Appendix A accompanying this article.

## 3. Results

From an original total of 21,990 articles, 28 were selected for inclusion in the review. Overall, 27 of the 28 articles included were in English; 1 article [24] was retrieved in Brazilian Portuguese and translated using the Google Cloud AI translation service. Figure 1 presents the PRISMA flow chart detailing the systematic literature search. Descriptive statistics were used to summarize the characteristics of the included studies. The in-depth analysis of the included studies is presented according to the Patterns, Advances, Gaps, Evidence for Practice, Research Recommendations (PAGER) framework [25]. This framework provides a standardized method for reporting the results of scoping reviews. The PRISMA guidelines were also used in this article [26], the checklist for which can be found in the Appendix A accompanying this article. A summary table of included studies is included below.

### 3.1. Study Characteristics

A summary of each article is provided in Table 1. The study characteristics of each article is included in Table 2. Graphs illustrating the years of publication and country contexts of included studies are presented in Figure 2 and Figure 3. 

#### 3.1.1. Year of Publication

Most of the included studies were published in 2022 (n = 12) [27,28,29,31,34,35,40,43,44,50,51,52], followed by 2019 (n = 5) [24,32,38,39,53]. The oldest study was published in 2013 [36]. The small number of included studies published in 2023 (n = 2) [30,45] likely reflects the last search date of this review (April 2023). The overall positive trend in publications over time suggests an increased awareness and interest in the economic impact of childhood rare diseases.

#### 3.1.2. Included Countries

The most common country featured in the included studies was the United States (n = 11) [27,28,32,35,38,40,41,42,44,45,52], followed by the United Kingdom (n = 8) [28,29,35,39,41,42,44,51]. Spain was included in seven studies [28,37,46,49,50,51,53]. Germany [28,29,41,42,44,51] and Italy [28,33,41,42,43,51] were included in six studies each, and France was included in five studies [28,29,30,44,51]. Canada [28,35], Mexico [28,50], the Netherlands [35,36], Australia [34,47], China [31,48], and Brazil [24,28] were discussed in two studies each. The countries included in one study each were Sweden, Ireland, Georgia, Russia, Japan, South Korea, Poland, and Argentina [28]. This geographical spread suggests a heavy bias towards European and Western contexts, as the only three studies included countries in Asia, and two studies include countries in Latin America [24,28].

#### 3.1.3. Study Design

Most of the studies (n = 18) included in the review were cross-sectional studies with a burden-of-illness focus [24,27,28,29,30,32,33,36,37,40,41,42,45,47,48,49,50,53]. Three of the studies were qualitative research [35,38,46], three were pilot studies [34,39,52], and another three were secondary analyses of data [31,44,51]. The remaining study was a cohort study [43].

Excluding the secondary analyses of data (n = 3), 22 of the included studies used quantitative methods for data collection. Validated scales for data collection were used in nine of the included studies [28,30,37,40,41,42,48,49,50]. These studies were predominantly assessing the burden of illness, and therefore included non-economic-specific validated scales that often included functional and psychological measures. The validated scales used included the FOP Physical Function Questionnaire (FOP-PF) (n = 1) [28]; the Epidermolysis Bullosa Burden of Disease (EB-BoD) (n = 2) [30,33]; the Parental Needs Scale for Rare Disease (n = 2) [39,47]; the Patient-Reported Mobility Assessment (PRMA) (n = 1) [28]; the EuroQoL EQ-5D (n = 4) [28,37,41,42]; the Patient-Reported Outcomes Measurement Information System Global Health Scale (n = 1) [28]; the W-BQ12 Well-being Questionnaire (n = 1) [30]; the 36-item Short Form (SF-36) Health Survey (n = 2) [37,48]; Health Utilities lndex Mark (HUI) (n = 2) [37,41]; the Impact on Family Scale (IOF) [37]; the Zarit Caregiver Burden lnterview (ZCBI) (n = 5) [37,42,48,49,50]; the World Health Organization (WHO) Health Productivity Questionnaire (HPQ) (n = 1) [40]; the National Health and Nutrition Examination Survey (NHANES) (n = 1) [40]; a Visual Analogue Scale (n = 1) [42]; the 12-item Short Form (SF-12) Health Survey (n = 2) [40,42]; the Social Support Rating Scale (n = 1) [48]; the Pittsburgh Sleep Quality Index (PSQI) (n = 1) [48]; the Patient Health Questionnaire-15 (PHQ-15) (n = 2) [49,50]; the Barthel Index (n = 1) [49]; the Satisfaction with Life Scale (SWLS) (n = 2) [49,50] and the Care-related Quality of Life-7D instrument (CarerQol7D) (n = 2) [49,50]. In total, 10 studies developed new surveys for their data collection [24,27,29,30,34,36,43,45,52,53]. Conner et al. [32] used data collection methods adapted from the BURQOL-RD project. This project looked at calculating the socioeconomic impact of rare diseases from a societal perspective across European countries [54]. Thus, a large variety of measurement tools were used across studies, demonstrating a lack of consistency in the literature.

#### 3.1.4. Population

Most of the studies (n = 14) focused on children within the full age range of childhood (i.e., 0–18 years) [24,28,29,31,34,38,41,44,45,46,47,48,49,50]. Nine studies presented more limited age ranges; however, this varied between small ranges (e.g., children aged 12–15 years in the study by Bourrat et al. [30]) and large ranges (e.g., 1–18 years in the studies conducted by Conner et al. [32] and Khair and Pelentsov [39]). Six studies included adults in their study population but sufficiently differentiated the results so that only children’s data were extracted [28,29,32,38,40,49]. Four studies did not report the age ranges of the children with rare diseases in their study [27,33,51,53]. Mean ages of children with rare diseases were reported in 11 studies [30,31,32,33,34,40,41,42,44,51,53].

The rare diseases explored in the included studies were grouped according to classifications from Orphanet. Pelentsov et al. [47] represented 132 distinct rare diseases. Across the remaining studies, the most common disease groups studied were inborn errors of metabolism (n = 11) [24,29,31,32,34,36,38,40,44,48,53], followed by musculoskeletal disorders (n = 7) [24,28,37,41,42,49,50]. Rare respiratory illnesses were studied in three included studies [24,35,45]. Rare bleeding disorders [39,51], skin disorders [30,33], and neurodevelopmental disorders [27,46] were studied in two studies each, while rare renal disorders [52] and rare neurological disorders [43] were studied in one study each. It should be noted, however, that many of the rare diseases included in the included studies have multi-systemic effects, and therefore this categorization likely lacks nuance.

Mothers were the most common respondent caregivers in the studies that reported the gender/roles of respondents (n = 17) [24,27,29,30,35,38,39,41,42,44,45,46,47,49,50,52,53] The only study that had more fathers than mothers in the respondent population was the study conducted by De Stefano et al. [33].

#### 3.1.5. Other Characteristics

Patient organizations were utilized by researchers in 20 of the studies [27,28,29,30,31,32,34,35,37,38,40,41,42,43,45,47,48,49,50,52]. This was often for assistance in sample recruitment. Only four studies co-designed their research methods with patient organizations [30,34,43,52].

### 3.2. Patterns

#### 3.2.1. The More Severe/the Higher Dependence of the Child with the Rare Disease, the Higher the Financial Burden

Several of the included studies (n = 8) noted that families of children with higher levels of dependence or disease severity, often categorized by stages of disease progression, assessed dependence levels, or the presence of co-morbidities, experienced higher financial costs than those with children with lower levels of dependence or less-severe rare diseases [28,36,37,44,49,50,51,52].

In Al Mukaddam et al. [28], increasing mobility loss was associated with greater financial burden, due to an increased need for aids, assistive devices or living adaptations or formal care. Children with later stage Duchenne Muscular Dystrophy (DMD), exemplified by lower ambulatory ability, were associated with higher annual household economic burden than those with early ambulatory DMD [41]. Similar associations between increased disease severity and higher economic costs were noted for neuromuscular rare diseases, including DMD, in the study by Rodríguez et al. [49]. Purchasing assistive devices to support decreasing mobility due to disease progression was noted as a significant cause of costs for children with these conditions [42]. In the study by Eijgelshoven et al. [36], having a child with severe Phenylketonuria (PKU), classified as a daily allowance of ≤10 g natural protein, incurred substantially higher out-of-pocket costs for caregivers than mild PKU (daily allowance of natural protein >10 g), as low-protein food products accounted for most expenses. The primary management strategy for PKU, which is characterized by a lack or deficiency in the enzyme phenylalanine hydroxylase, centers around protein diet control as phenylalanine occurs in natural protein foods [55]. Therefore, families must spend money on special low-protein foods to maintain their child’s health. Similarly, in a study of children with nephrotic syndrome, families of those that had progressed to end-stage kidney disease (ESKD) had increased out-of-pocket costs than those that had not progressed to ESKD [52]. Caregivers of children who underwent liver transplants due to Progressive Familial Intrahepatic Cholestasis (PFIC) reported more instances of presenteeism and absenteeism than caregivers of children with PFIC without a surgical history [44], which can influence caregivers’ financial burden. Similarly, De Stefano et al. [33] found that caregivers’ reduced working hours were directly correlated with the severity of their child’s Epidermolysis Bullosa, and that more severe forms of the condition were associated with higher economic burden for families. Children with severe hemophilia receiving an on-demand treatment regimen had higher costs compared to children with less-severe hemophilia in the study by Rodriguez-Santana et al. [51].

Caregivers of children with Duchenne Muscular Dystrophy (DMD) were six times more likely to experience a high financial burden due to comorbidities such as language and learning difficulties, obsessive–compulsive disorder, irritability, and sociability issues in Flores et al.’s study [37]. Children with DMD and learning difficulties were also six times more likely to need formal care (care delivered by a paid professional caregiver) [37], which has its own associated monthly expenditure.

In some cases, rare diseases with higher severity associated costs, such as direct care costs, repeated hospital visits and transport, are covered by state benefits. While the need for this type of support has been demonstrated in this section, it may exclude families with children with milder conditions, who may incur fewer costs but may also experience financial strain. Khair and Pelentsov [39] noted that 42.5% of respondents in their study reported financial burden due to their child’s rare disease. Therefore, while costs to families may rise with increased severity/dependence of their child’s rare disease, it is important to recognize that all families with children with rare diseases may be exposed to financial burden.

#### 3.2.2. Children with Rare Diseases Attend a High Variety of Healthcare Professionals for Health Management, Often with Associated Costs

Frequent visits to healthcare and allied health professionals were noted in 13 of the included studies [27,28,30,32,34,35,41,43,45,46,49,50,52].

Visits to doctors [28,34,41,49] and dentists/orthodontists [28,32,52] were frequently mentioned in the included studies; however, it was not always clear if these visits were covered by insurance or expensed by the family themselves. Conner et al. [32] noted visits to a genetic counsellor. However, this was one of the healthcare professionals’ visits with the lowest rate of full insurance coverage. Conner et al. [32] and Simon et al. [52] also noted visits to a chiropodist, optometrist, social worker, and behavioral therapist. Depending on the rare disease and health system, some children had access to healthcare professionals with specific expertise in their condition, as in the studies by Driessens et al. [35] and Nelson et al. [45], while some reported difficulties in accessing this type of specialty care [27].

Among the 31% of caregivers who chose to receive psychological care in the study by Bourrat et al. [30], 80% had to pay their own consultation fees. Cost was a barrier for nearly half of the respondents who did not receive this care [30].

Visits to allied health professionals were frequent and often resulted in out-of-pocket costs for families [34]. Commonly cited rehabilitative therapies included physiotherapy [41,46,49], physical therapy [27,32,43], psychologists [32,49], occupational therapy [27,32,43,46], dietician [35], speech therapy [27,32,43,46,49], music therapy [21], and alternative therapies such as aqua therapy, equine therapy, or hippotherapy [27,49]. These visits were often costly for the family, accounting for most of the household income expenditure in the study by Rodríguez et al. [49], although they were covered by insurance in other studies [32,43]. Caregivers spent approximately the same proportion (mid-range 28% (Spain)–33% (Mexico)) of the family income on visits to healthcare professionals as they did on assistive technology for their child in the study by Rodríguez et al. [50].

#### 3.2.3. Health-Related Consumable Items Can Account for Significant Expenditure but Are Not Always Covered by Health Insurance

Substantial out-of-pocket costs were noted for health-related consumable items in seven of the included studies [30,34,36,43,46,52,53]. These items were finite and purchased for requirements of the child’s rare disease, and included dietary supplements [43], diet-specific food products (e.g., low protein foods for PKU) [36,52,53], cleansing products and moisturizing creams or emollients [30], continence aids [34], sanitary supplies [46], special clothes [30], and complementary or alternative medicines [30]. Medical consumables include dressings, tubing, and syringes [34]. The costs of these products were often not covered by insurance or regulated as they are not considered medical costs, leaving families the burden of covering these expenses [30,46]. Financial cost in Simon et al.’s study [52] was one barrier to accessing diet-specific food products.

#### 3.2.4. Many Families of Children with Rare Diseases Must Pay Out-of-Pocket Payments Regardless of Insurance Status

As previously discussed, out-of-pocket costs were experienced by families in many of the included studies (n = 11) [24,28,30,31,34,36,37,41,43,46,52]. Items responsible for these costs varied between the studies. Visits to healthcare professionals and expenditure on aids, assistive devices or adaptations constituted the bulk of these costs in the study by Al Mukaddam et al. [28]. In Bourrat et al.’s study [30], families of children with Epidermolysis Bullosa mainly experienced out-of-pocket costs for transport, housing adaptations and disease-specific expenses, such as cleansing products, emollients, and moisturizing creams. Expenditure increased with the increasing age of the child and varied between the different Epidermolysis Bullosa subtypes [30]. Disease-specific expenses were also a feature in Eijgelshoven et al.’s study [36], as diet supplements, food, and equipment, among other impacts specific to PKU, were the main drivers of out-of-pocket expenditure. However, as discussed in a previous section, these out-of-pocket expenses were more influenced by the severity of the disease than the child’s age.

The variety of out-of-pocket cost items in each of the studies are affected by insurance types [52]. Travel and accommodation to access care, as well as the associated consultation fees for medical and allied health specialists, non-prescription medicines, and consumables not covered by health insurance, accounted for most out-of-pocket expenses for families of children with tuberous sclerosis and mitochondrial disorders in Deverell et al.’s study [34]. The costs of medicines and formal care for children with DMD that the Spanish healthcare system did not cover were components of out-of-pocket costs in the study by Flores et al. [37]. For families of children with DMD in Germany, Italy, the United States, and the United Kingdom, out-of-pocket costs depended on the different health systems and the characteristics of the received healthcare. However, families in the United States experienced significantly higher out-of-pocket costs than in other countries [41]. This variation in insurance coverage varies between countries, but also within populations, as the families in Lo Barco et al.’s study [43] experienced different levels of rehabilitation cost coverage; some had all hours covered, while others had to bear all costs of this treatment. According to Palacios-Ceña et al. [46], all expenses not directly related to hospitalization are covered by families, including supportive therapies and consumable products.

Chen and Dong [31] modelled insurance reimbursement schemes for families of Pompe Disease patients using cross-sectional rare disease survey data. They found that approximately 96% of families of children with Pompe Disease experience catastrophic health expenditures (defined in the study as out-of-pocket costs exceeding 10% of annual family income) before any reimbursement from insurance. Extrapolating these data, they found that if these families were reimbursed and direct non-medical costs were included, 87.5% would still face catastrophic health expenditures due to their child’s rare disease [31]. Catastrophic health expenditure was also experienced for 12% of families of children with mucopolysaccharidoses and 31% of families of children with osteogenesis imperfecta in the study by Pinto et al. [24]. This illustrates the significance of out-of-pocket costs for this population.

### 3.3. Advances

#### Developments of Validated Scales/Measurement Instruments Specific to Rare Diseases

Five studies utilized rare disease-specific validated scales when collecting data on the impact of rare diseases on children and their families, including their functional status and quality of life. These scales also included questions on the economic impact families faced due to their child’s rare disease. The instruments used include the FOP Physical Function Questionnaire (FOP-PF) (n = 1) [28], the Epidermolysis Bullosa Burden of Disease (EB-BoD) (n = 2) [30,33], and the Parental Needs Scale for Rare Disease (n = 2) [39,47].

The development of scales specifically to measure the impacts of rare diseases may advance research in this field. The complexity of rare diseases means combining multiple measurement tools to gather relevant data may be necessary, which may lead to lengthy surveys. This may prevent uptake amongst participants and lead to non-responses. Lengthy questionnaires have been demonstrated to significantly influence participant response rates [56]. Using tailor-made measurement tools for rare diseases, researchers can identify and measure rare disease impacts efficiently and effectively. Scales such as these can assist in illustrating the overall disease burden of rare diseases, and assist in the intervention evaluations, improving health and supportive care to the population through an improved evidence base [57].

### 3.4. Gaps

#### 3.4.1. The Impact of Productivity Loss (Presenteeism) and Informal Care Are Substantial but Not Always Quantified

While most of the included articles discussed productivity loss in terms of presenteeism, absenteeism and change in work status/employment among caregivers (n = 20) [24,29,30,32,33,34,35,36,37,38,40,41,42,44,46,47,48,49,52,53], few quantified the impact of these factors on family income.

Respondents across the included studies reported reducing their working hours or quitting their jobs altogether (n = 15) [24,29,32,33,34,35,36,37,38,40,41,44,46,49,53]. In the study by Tejada-Ortigosa et al. [53], parents reported a lack of support from their employer in achieving balance between working and caring for their child with a rare disease. Caregivers also had to decline promotions or job opportunities [40,52], cease education [47,52], take time off or leaves of absence [30] and/or change jobs [35,36,37,40]. Typically, respondents were asked to report productivity loss in hours or days spent performing caring duties instead of lost work hours or the financial implications of these losses.

Losses in caregivers’ leisure time due to informal care needs were measured in the studies by Landfeldt et al. [41], Landfeldt et al. [42], and Rodríguez et al. [49]. However, these impacts were often measured by frequency or self-reported incidence, and rarely quantified to understand how much these factors cost. Some studies evaluated the hours caregivers spent on care but did not quantify the lost wages or cost of informal care for these hours [24,31,41,51]. Only three studies quantified productivity loss: Landfeldt et al. [41], Qi et al. [48], and Rodríguez et al. [49]. In Landfeldt et al.’s study [41], hours of paid informal care were calculated using the Human Capital approach. Figures ranged from an average of EUR 13,160 to EUR 18,530 annually across Germany, Italy, the United Kingdom, and the United States, and were often the most significant components of the cost of illness of DMD to families. Qi et al. [48] determined the cost of productivity loss to caregivers due to caring by calculating their daily lost wages. Annually, caring for a child with Gaucher Disease costs caregivers USD 1980 USD in productivity loss alone. The cost of informal care to families of children with neuromuscular disorders was quantified by Rodríguez et al. [49] through the replacement method, using the minimum hourly rate for live-in carers in Spain. The total costs varied according to disability level, with mild dependence costing an average of EUR 6563.84 and severe dependence an average of EUR 17,922.81 annually.

#### 3.4.2. The Economic Impact of Sourcing Diagnostic Tests and Treatments Has Only Been Explored in a Few Studies

The lack of availability and affordability of rare disease treatments is a common challenge for patients with rare diseases and their families, and families’ ability to access treatment was mentioned in seven of the included studies. Navigating insurance coverage and financial challenges to accessing appropriate treatment were noted as key challenges for families of children with rare diseases in the studies by Palacios-Ceña et al. [46], Gerstein et al. [38], and Ak et al. [27].

In Qi et al.’s study [48], unaffordability of therapeutic drugs was reported by all respondents. Taking medicine at home accounted for 61.3% of the total costs of caring for a child with Gaucher Disease (calculated as EUR 48,771 USD), with medical tests accounting for 3.4%. Transplant-associated costs were significant for families of children with Nephrotic Syndrome in Simon et al.’s study [52], costing approximately USD 1800. In Conner et al.’s study [32], diagnostic tests were only fully covered for 73–84% of respondents, necessitating some families to pay out of pocket for procedures such as blood tests, radiology, echocardiograms, electrocardiograms, and urinalysis. Similarly, rehabilitation costs for children with SYNGAP1 developmental and epileptic encephalopathy were not always fully covered in Lo Barco et al.’s study [43], meaning that some families had to take full responsibility for these costs.

In Palacios-Ceña et al.’s study [46], the inequality in access to genetic diagnosis was noted, which widened health inequalities between families of children with rare diseases. Families who did not live in cities where this diagnostic method was covered under public health services had to pay out of pocket, which in some cases, was a barrier as they did not have the funds [46].

#### 3.4.3. Few Rare Disease Groups Were Explored, with No Cardiac or Oncological Rare Diseases Included

There are over 6000–10,000 rare diseases [3,5], which can be classified according to many different categorizations. Classifying rare diseases by body systems, as used in this review, can be difficult as many rare diseases are multi-systemic, and thus may need to be more neatly fit into a single category. However, there is a noted absence in this review of any rare cardiovascular or oncological diseases in the included studies.

Cardiovascular diseases in children are rare but often have significant impacts on health status. These diseases, including pediatric cardiomyopathy, can lead to heart failure and necessitate heart transplantation [58]. As previously discussed, having a child with a rare disease undergo organ transplantation can indirectly impact family finances through reducing caregivers’ productivity, causing higher levels of presenteeism and absenteeism [44]. Cancer in childhood is rare and is associated with significant economic impacts on families. A recent report by the Irish Cancer Society found that the associated direct healthcare, direct non-healthcare, and indirect costs of having child with cancer result in an increase of EUR 2250 in family expenses after diagnosis, along with an average monthly loss of EUR 1280 in income due to absenteeism, presenteeism, and changes in work status [59].

The lack of studies regarding cardiovascular or oncological diseases included in this review may result from the search strategy. Cardiovascular and oncological diseases are established and well-known fields, and therefore may only sometimes utilize their rare definition. It may also reflect the siloes of evidence in the rare disease field, along with a lack of acknowledgement of the expected impacts faced by families of children with rare diseases regardless of the disease group.

Defining rare diseases by their incidence, e.g., rare versus ultra-rare, was not utilized in this review. This was to reflect the differences in the countries included as part of the inclusion criteria. For example, in a study by Castillon, Chang, and Moride [60], which looked at the global incidence rates of Gaucher disease, there was a variation between 0.45 and 25.0 per 100,000 people across Europe, North America, and the Asia-Pacific region. This reflects the highly genetic nature of rare diseases and would have necessitated this review taking a specific country’s perspective in defining which of the rare diseases included were ‘common’ rare or ultra rare.

## 4. Discussion

This review aimed to synthesize knowledge around the economic impacts of having a child with a rare disease for families, as highlighted by the identified patterns, advances, and gaps. These impacts, such as income, unemployment, and job security, are just some of the social determinants of health that can have significant influence on all health outcomes [61]. However, the complexity of rare diseases can compound these influences and affect not just these patients but their families, as the challenges of caring for someone with a rare disease can impact their own health as well. Therefore, addressing these impacts can improve the quality of life of rare disease patients and their families [62].

Across the three domains of expenses, direct healthcare, direct non-healthcare and indirect costs, the main categories of expenditure related to visits to healthcare and allied healthcare professionals, health-related consumable items, informal care and productivity loss (presenteeism and absenteeism). The review found a link between the severity/dependence of the child with the rare disease and the cost burden for families. It was also found that these families often have high rates of out-of-pocket expenditure. However, it is important to note the need for more standardization to measure this economic burden. Few studies used validated tools to measure economic impacts, and fewer still calculated the cost burden using economic approaches such as the Human Capital method. This is likely due to the dominance of cross-sectional study designs that rarely focus on the economic impact on families. Furthermore, the studies were predominantly based in Western and developed country contexts and had notable gaps in the rare disease groups covered.

### 4.1. Evidence for Practice and Research Recommendations

This review did not focus on clinical interventions or programs, so little evidence for practice was retrieved. Most studies on the economic impact of rare diseases focus on societal perspectives through cost-of-illness studies [16]. This study adopted a different approach and focused on costs from the perspective of families. Though often limited, the family perspective of the economic burden of rare diseases is important to understand the costs, whether direct medical, non-medical, or indirect, that the rare disease population experiences. These perspectives can be considered collectively to define rare diseases’ impact and generate a societal view of the economic burden of rare diseases using a person-centered approach [16]. The studies included in this review highlight the importance of financial support for families of children with rare diseases, due to the wide-ranging and often significant costs associated with these conditions. While state financial assistance and insurance coverage is present, it is limited and unlikely to cover areas of significant daily expenditure for families. These impacts can considerably impact the mental wellbeing and quality of life of caregivers. Therefore, widening coverage and ensuring families’ financial wellbeing can enhance the provision of holistic care for this population. However, it is important to note that making inferences about the coverage level of insurance or state support across the included studies is difficult, as not all discussed this topic and those that explored it in varying depths.

Healthcare professionals must be aware of the significant economic impact experienced by families of children with rare diseases. Visits to healthcare professionals account for a substantial expenditure for this population. Still, other costs not associated with healthcare may have sizable impacts on their lives and the care they access. For example, symptoms or increased comorbidities experienced by children with rare diseases not necessarily captured by ICD-10 coding may be associated with costs not visible to healthcare professionals. Improved care or quality of life may be restricted by families’ ability to afford assistive devices or housing adaptations, which can be extremely expensive. Improving healthcare professionals’ knowledge of these impacts will contribute to their understanding and care of rare disease patients. The creation of dedicated rare disease centers and integration of European Reference Networks (ERNs) into national systems (as performed by the JARDIN project [63] may contribute to this increase in knowledge, as they provide focused resources to rare disease populations.

### 4.2. Policy Implications

Understanding the economic costs of living with a rare disease for families can contribute to the development of sensitive and comprehensive policy to address the challenges they face. The impacts of rare diseases on socioeconomic factors and determinants of health such as household income, appropriate accommodation and participation in society all fall under governmental policy remits. Furthermore, at a closer level, the policies of healthcare institutions may be influenced by greater understanding of these impacts to provide more responsive care to patients. For example, the introduction of virtual rare disease clinics, as exemplified by the telehealth nurse-led ‘Nurse Navigator Program’ at the Rare Care Centre at Perth Children’s Hospital [64]. Services like these may alleviate the costs experienced by families who would otherwise have to pay for travel, accommodation, parking and food during an in-person visit to the hospital. While it is beyond the scope of this review to suggest particular policy initiatives, the authors advocate for more research on the economic impact of rare diseases to influence policy and improve the quality of life for families of children with these conditions.

### 4.3. Consultation

Consultation is an optional step within Arksey and O’Malley’s [12] scoping review framework. However, there is no current consensus on how to approach this [65]. This review is part of a larger PhD project conducted by the first author. Prior to commencing this research, the review protocol was presented to the public and patient involvement (PPI) representative group associated with this project, who reviewed, suggested edits, and approved the review questions. This group is made up of parents of children and young people with rare diseases, who have experience of caring for these children and the associated economic impacts on the family. Following the completion of this scoping review, the findings were presented to these same representatives, who validated the results and provided their feedback and recommendations for future areas of focus within the PhD project. The findings were also presented to PPI members of the All-Ireland Rare Disease Interdisciplinary Research Network (RAiN), for their comments and feedback.

### 4.4. Limitations

This review has several limitations. Firstly, only peer-reviewed literature was included in the review, resulting in the omission of grey literature. Furthermore, the study types included in this review were largely cross-sectional in nature and often did not focus on economic impacts alone. This is likely due to the review’s focus on caregiver-reported impacts. Few studies examined these impacts in isolation, often considering them with other domains, including psychological wellbeing or quality of life, therefore minimizing the amount of economic data to include in the review.

This review focuses on highlighting the types of economic costs experienced by families of children with rare diseases. It does not seek to compare the extent of the burden of these costs between rare diseases or country contexts, as the lack of validated, universally accepted assessment tools makes this comparison impossible. However, understanding the categories of frequently assessed costs across diseases and countries, as this review provides, may contribute to the development of tools that may enable this comparison in the future.

Due to the variations in studies retrieved, and the need for a standardized quality assessment tool for cost-of-illness studies, a quality analysis was not performed. However, this is not a required component of a scoping review, and thus is an accepted limitation.

The global nature of the review means that the studies were all conducted within the context of a specific health system; however, though information was gathered during data charting, it ultimately had little influence on the impacts reported as the review did not aim to quantify the financial costs. Where relevant, different insurance coverage and health system limitations were discussed; however, the type of health system the research was based on was rarely discussed in the included studies. Nevertheless, the included studies revealed the limited geographical distribution, as most of the studies focused on Western contexts. This may have also been influenced by the English language inclusion criterion but reduces the generalizability of the results of this review to other contexts. It may also reflect the countries in which there are dedicated centers or networks for rare diseases, as rare disease care has yet to be integrated into all national health systems; therefore, these diseases may remain hidden or under-researched in many countries.

A key aspect of this review’s inclusion and exclusion criteria was the differentiation between the adult and child populations. Only studies that clearly differentiated the impacts between these age groups were included. However, one study [28], although distinguishing between these populations regarding the physical and quality of life impacts of fibrodysplasia ossificans progressive (FOP), presented the economic impacts for both these populations combined. The study was retained because the statistical analysis was performed separately for each age group. A consultation with stakeholders on the results of the review was not performed. This is noted as optional in the PAGER framework outlined by Bradbury-Jones et al. [25].

## 5. Conclusions

This review explores the evidence surrounding the economic impact of rare diseases on children and their families. Most of the evidence is derived from the burden of illness studies, which vary in the detail they provide about these economic impacts. Nonetheless, families incur substantial expenses related to their child’s rare disease. Significant areas of expenditure include visits to healthcare professionals and health-related consumable products. Additionally, families face different levels of out-of-pocket expenses and insurance coverage depending on their country’s healthcare system. The severity of the disease and the child’s level of dependence are significant predictors of family spending. However, understanding these costs is the first step towards developing interventions and support systems that can enhance these families’ financial and overall well-being.

## Figures and Tables

**Figure 1 healthcare-12-02578-f001:**
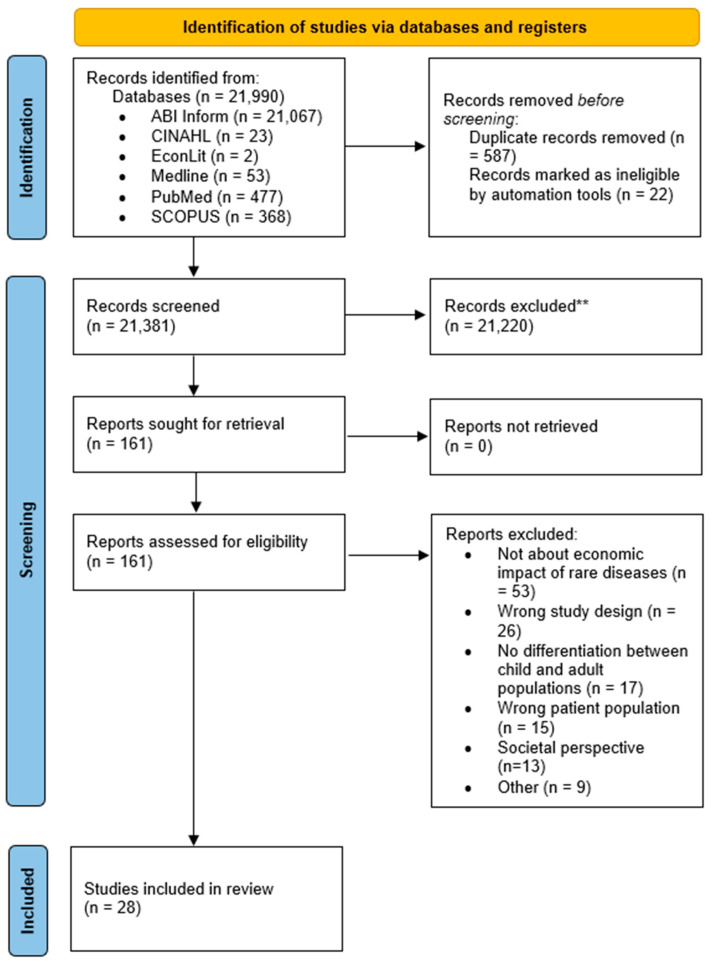
PRISMA flow chart of literature search, adapted from Page et al. [26].

**Figure 2 healthcare-12-02578-f002:**
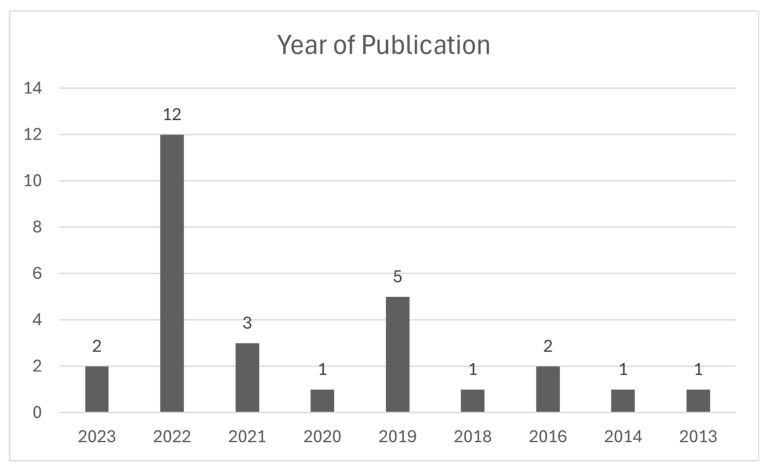
Year of publication of included studies.

**Figure 3 healthcare-12-02578-f003:**
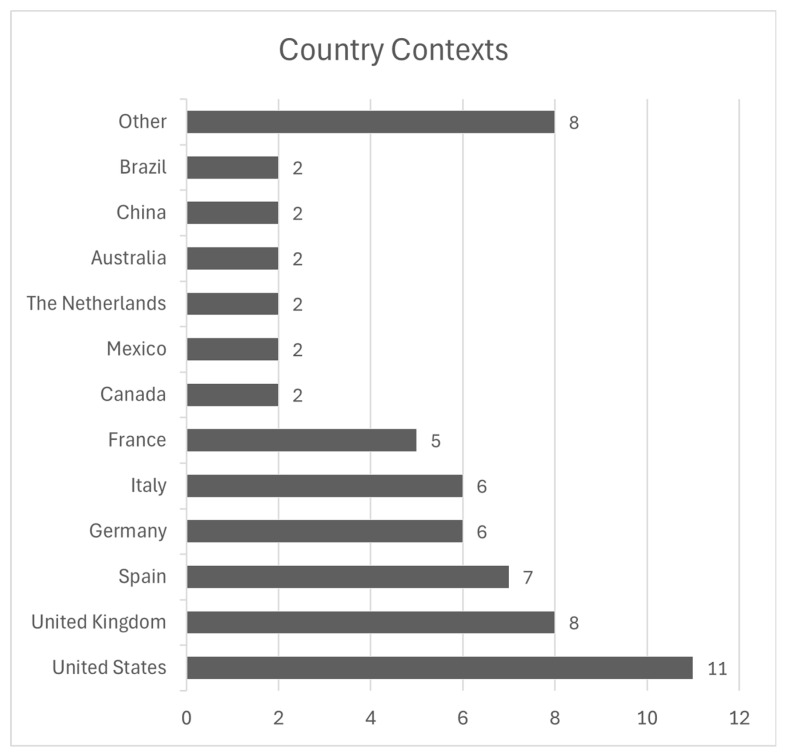
Country contexts of included studies.

**Table 1 healthcare-12-02578-t001:** Article summary table.

Author	Rare Disease	Economic Impact Category	Economic Impacts Measured
Ak et al., (2022) [27]	MECP2 Duplication Syndrome	General	Financial wellbeing, insurance and financial difficulties
Al Mukaddam et al., (2022) [28]	Fibrodysplasia Ossificans Progressiva	Direct medical, indirect	Living adaptations, assistive devices, absenteeism, career decision-making
Bashorum et al., (2022) [29]	Fabry disease	Indirect	Absenteeism
Bourrat et al., (2023) [30]	Epidermolysis Bullosa	Direct medical, indirect	Employment status, absenteeism, presenteeism, out-of-pocket expenditure, health-related consumable items
Chen and Dong, (2023) [31]	Pompe disease	Direct medical, direct non-medical	Out-of-pocket expenditure
Conner et al., (2019) [32]	Mucopolysaccharidosis Type I	Direct medical, indirect	Treatment costs, visits to healthcare professionals, absenteeism, presenteeism
De Stefano et al., (2021) [33]	Epidermolysis Bullosa	General, indirect	‘Economic impact’, presenteeism, employment status
Deverell et al., (2021) [34]	Tuberous Sclerosis and Mitochondrial disorders	Direct medical, direct non-medical, indirect	Health-related out-of-pocket expenditure (including travel for medical care and accommodation, visits to healthcare professionals, alternative medicines, treatments, medical equipment, living adaptations, health-related consumable items), presenteeism, employment status
Driessens et al., (2022) [35]	Primary Ciliary Dyskinesia	Direct medical, indirect	Visits to healthcare professionals, absenteeism, presenteeism, employment status
Eijgelshoven et al., (2013) [36]	Phenylketonuria	Direct medical, direct non-medical, indirect	Out-of-pocket expenditure on health-related consumable items, extra holiday luggage for equipment, postage cost for blood tests, PKU-related events, employment status
Flores et al., (2020) [37]	Duchenne Muscular Dystrophy	General, direct medical, indirect	Medical costs, financial burden, presenteeism, absenteeism, employment status
Gerstein et al., (2019) [38]	Urea Cycle disorders	Direct medical, direct non-medical, indirect	Cost of care, insurance coverage, health-related consumable items, treatments, travel for medical care, presenteeism
Khair and Pelentsov, (2019) [39]	Bleeding disorders	General, direct medical, direct non-medical	Financial difficulties, medical costs/treatments, medical equipment, health-related consumable items, respite care
Kruger et al., (2022) [40]	Long-Chain Fatty Acid Oxidation disorders	Indirect	Presenteeism, absenteeism, employment status
Landfeldt et al., (2014) [41]	Duchenne Muscular Dystrophy	Direct medical, direct non-medical, indirect	Assistive devices, medications, diagnostics and treatment, hospitalization, visits to healthcare professionals, non-medical community services, informal care, presenteeism, employment status, out-of-pocket expenditures
Landfeldt et al., (2016) [42]	Duchenne Muscular Dystrophy	General, indirect	Household cost burden, informal care
Lo Barco et al., (2022) [43]	SYNGAP1-developmental and epileptic encephalopathy	Direct medical, direct non-medical	Visits to healthcare professionals, travel for medical care, medications, health-related consumable items
Mighiu et al., (2022) [44]	Progressive familial intrahepatic cholestasis (PFIC)	Indirect	Absenteeism, presenteeism, career decision-making
Nelson et al., (2023) [45]	Pulmonary hypertension	Direct medical, direct non-medical	Visits to healthcare professionals, treatment costs, travel for medical care, relocation
Palacios-Ceña et al., (2018) [46]	Rett syndrome	Direct medical, indirect	Presenteeism, treatments, visits to healthcare professionals, health-related consumable items, living adaptations, diagnostic tests
Pelentsov et al., (2016a) [47]	Rare diseases (n = 132)	General, direct medical, indirect	Medical care and therapy, presenteeism, employment status, parents coping financially
Pinto et al., (2019) [24]	Cystic Fibrosis, Mucopolysaccharidosis and Osteogenesis Imperfecta	Direct medical, direct non-medical, indirect	Diagnostic tests, hospitalization, financial coping strategies (loans, selling assets), Presenteeism, employment status
Qi et al., (2021) [48]	Gaucher disease	Direct medical, direct non-medical, indirect	Medications, hospitalization, outpatient visits, diagnostic and medical tests, travel for medical care and accommodation, meals, health-related consumable items, productivity loss
Rodríguez et al., (2021) [49]	Neuromuscular Disease	Direct medical, direct non-medical, indirect	Visits to healthcare professionals, assistive devices, education support, health-related consumable items, informal care, employment status
Rodríguez et al., (2022) [50]	Duchenne Muscular Dystrophy	Direct medical, indirect	Visits to healthcare professionals, assistive devices, hours spent on care
Rodriguez-Santana et al., (2022) [51]	Hemophilia	Direct medical, direct non-medical, indirect	Healthcare resource use, medical equipment, formal care, diminished work productivity
Simon et al., (2022) [52]	Nephrotic syndrome	Direct medical, direct non-medical, indirect	Medication, health-related consumable items, hospitalization, visits to healthcare professionals, treatment, insurance, hours spent meal-prepping, productivity loss
Tejada-Ortigosa et al., (2019) [53]	Metabolic rare diseases	Direct medical, indirect	Medication, health-related consumable items, treatments, presenteeism, employment status

**Table 2 healthcare-12-02578-t002:** Study characteristics.

Article	Country	Study Design	Sample Pediatric Population	Original Scale Developed (If Any)	Validated Scales Used (If Any)	Use of Patient Organisation
Ak et al., (2022) [27]	USA	Cross-sectional study	85 caregivers of children (0–17 years) with MECP2 duplication syndrome (MDS)	Yes	N/A	Yes
Al Mukaddam et al., (2022) [28]	Argentina; Canada; Germany; Spain; Italy; France; Brazil; Japan; Mexico; Poland; Russia; South Korea; Sweden; United Kingdom; USA.	Cross-sectional study	Patients (0–24 years) (26.6% of total sample) and caregivers of children with fibrodysplasia ossificans progressive (FOP)	N/A	Patient-Reported Mobility Assessment (PRMA); the FOP Physical Function Questionnaire (FOP-PF); the EQ-5D-5; the Patient-Reported Outcomes Measurement Information System Global Health Scale	Yes
Bashorum et al., (2022) [29]	Germany; France; United Kingdom.	Cross-sectional study	A total of 14 patients with Fabry disease, 14 caregivers and five HCPs	Yes	N/A	Yes
Bourrat et al., (2023) [30]	France	Cross-sectional study	77 parents of children with different epidermolysis bullosa types and disease severity (40 female, 37 male; mean age 7.5 years)	Yes	W-BQ12 Well-being Questionnaire; Epidermolysis Bullosa Burden of Disease (EB-BoD)	Yes
Chen and Dong, (2023) [31]	China	Secondary Analysis of Data	24 pediatric patients with Pompe Disease	Yes	N/A	Yes
Conner et al., (2019) [32]	USA	Cross-sectional study	32 parents living in the US with children with severe mucopolysaccharidosis type I who were younger than 18 years of age	N/A	Adapted data collection tools from the BURQOL Project	Yes
De Stefano et al., (2021) [33]	Italy	Cross-sectional study	50 parents of EB children and adolescents (26 male, 24 female; mean age 9.7+/−5.5 years).	N/A	EB Burden of Disease (EB-BoD)	No
Deverell et al., (2021) [34]	Australia	Pilot Study	13 families with 15 children; median age 7 years (range: 1–12); 5 with mitochondrial disorders, 10 with tuberous sclerosis	Yes	N/A	Yes
Driessens et al., (2022) [35]	Canada; Georgia; Ireland; The Netherlands; United Kingdom; USA.	Qualitative research	18 mothers and 6 fathers of children under 6 years with Primary ciliary dyskinesia (PCD)	N/A	N/A	Yes
Eijgelshoven et al., (2013) [36]	The Netherlands	Cross-sectional study	24 caregivers of pediatric patients with PKU (median age 11 years; 9 mild and 12 severe disease severity)	Yes	N/A	No
Flores et al., (2020) [37]	Spain	Cross-sectional study	36 families looking after 38 DMD patients (34 patients <16 years old)	N/A	36-item Short Form (SF-36); Health Utilities lndex Mark (HUI); EuroQoL EQ-5D; Impact on Family Scale (IOF); Zarit Caregiver Burden lnterview (ZCBI)	Yes
Gerstein et al., (2019) [38]	USA	Qualitative research	Parents (n = 35) and providers (n = 26) of children diagnosed with urea cycle disorders	N/A	N/A	Yes
Khair and Pelentsov, (2019) [39]	United Kingdom	Pilot Study	231 parents of children with a bleeding disorder.	N/A	Parental Needs Scale for Rare Disease	No
Kruger et al., (2022) [40]	USA	Cross-sectional study	30 caregivers reporting on 37 individuals with long-chain fatty acid oxidation disorders.	Yes	The World Health Organization (WHO) Health Productivity Questionnaire (HPQ); National Health and Nutrition Examination Survey (NHANES) Physical Activity Questionnaire (PAQ); 12-Item Short Form of the Medical Outcomes Survey (SF-12)	Yes
Landfeldt et al., (2014) [41]	Germany; Italy; United Kingdom; USA.	Cross-sectional study	770 patient–caregiver pairs (pediatric age range 8–17 years) with Duchenne Muscular Dystrophy	N/A	Health Utilities Index and EuroQolEQ-5D instrument	Yes
Landfeldt et al., (2016) [42]	Germany; Italy; United Kingdom; USA.	Cross-sectional study	770 caregivers of children with Duchenne Muscular Dystrophy (mean age of 14 years; median 12 years; interquartile range 9–17 years)	N/A	EuroQol EQ-5D; Visual Analogue Scale (VAS); SF-12 Health Survey; Zarit Caregiver Burden Interview (ZBI)	Yes
Lo Barco et al., (2022) [43]	Italy	Cohort study	Caregivers of 13 children and adolescents with SYNGAP1-developmental and epileptic encephalopathy (SYNGAP1-DEE)	Yes	N/A	Yes
Mighiu et al., (2022) [44]	Germany; France; United Kingdom; USA.	Secondary Analysis of Data	22 caregivers of patients with progressive familial intrahepatic cholestasis (average age 8.2 years)	N/A	N/A	No
Nelson et al., (2023) [45]	USA	Cross-sectional study	139 parents of a child living with pediatric pulmonary hypertension	Yes	N/A	Facebook group
Palacios-Ceña et al., (2018) [46]	Spain	Qualitative research	31 caregivers of children with Rett syndrome	N/A	N/A	No
Pelentsov et al., (2016a) [47]	Australia	Cross-sectional study	301 parents of children with 132 distinct rare diseases from Australia and New Zealand	Yes	Parental Needs Survey	Yes
Pinto et al., (2019) [24]	Brazil	Cross-sectional study	Caregivers from 99 families with children (total n = 106) with cystic fibrosis (n = 62), mucopolysaccharidosis (n = 16) and osteogenesis imperfecta (n = 28)	Yes	N/A	No
Qi et al., (2021) [48]	China	Cross-sectional study	40 patients and their 49 caregivers with Gaucher Disease (pediatric patients 0–14 years n = 38)	N/A	Zarit Burden Inventory; Social Support Rating Scale; Pittsburgh Sleep Quality Index [PSQI]; Short Form Health Survey (SF-36)	Yes
Rodríguez et al., (2021) [49]	Mexico; Spain.	Cross-sectional study	34 caregivers from Mexico and 40 from Spain of children with Duchenne Muscular Dystrophy (DMD)	N/A	Patient Health Questionnaire-15 (PHQ-15); Zarit Caregiver Burden Scale; Satisfaction with Life Scale (SWLS); CarerQol7D	Yes
Rodríguez et al., (2022) [50]	Spain	Cross-sectional study	110 parents of children with neuromuscular disease	N/A	Patient Health Questionnaire-15 (PHQ-15); Zarit Caregiver Burden Scale; Satisfaction with Life Scale (SWLS); CarerQol7D; Barthel Index	Yes
Rodriguez-Santana et al., (2022) [51]	Germany; Frane; Italy; Spain; United Kingdom.	Secondary Analysis of Data	794 children and adolescents with moderate and severe hemophilia A or B (mean age 10.5 years)	N/A	N/A	No
Simon et al., (2022) [52]	USA	Pilot Study	17 caregivers of children with primary FSGS, minimal change disease, IgM nephropathy, membranous nephropathy, or childhood-onset idiopathic nephrotic syndrome for at least 1 year, residing in the United States	Yes	N/A	Yes
Tejada-Ortigosa et al., (2019) [53]	Spain	Cross-sectional study	65 parents and/or legal guardians of pediatric patients with different inborn errors of metabolism	Yes	N/A	No

## Data Availability

No new data were created or analyzed in this study.

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
