# Peer review of "Caregiver-Reported Economic Impacts of Pediatric Rare Diseases—A Scoping Review"

_healthcare, 2024, doi:10.3390/healthcare12242578_

Round 1

Reviewer 1 Report

Comments and Suggestions for Authors

My decision is to reject this manuscript.

Summary of the Study

The article provides a scoping review on the economic impact of pediatric rare diseases on caregivers, specifically focusing on direct medical, direct non-medical and indirect costs. The authors employ the PAGER framework to map and summarize the literature, synthesizing data from 28 studies. This paper highlights areas of high economic burden such as healthcare visits, medication costs, presenteeism and informal care giving while identifying significant gaps in the literature, such as the limited focus on ultra-rare diseases and economic costing methods.

Strengths Comprehensive Scope

The article successfully covers a wide range of costs associated with pediatric rare diseases, contributing to the under-researched area of the economic burden on families.

Framework Application: The use of the PAGER framework is commendable as it provides a structured and systematic approach to reporting scoping reviews. This adds clarity to the results. Identification of Gaps: The paper highlights key research gaps in the field such as the lack of distinction between rare and ultra-rare diseases which is crucial for future research. The focus on the need for better economic evaluation tools is insightful.

Weaknesses and Areas for Improvement

Limited Geographical Representation: The studies predominantly focus on Western contexts (Europe and the United States) with very few addressing rare diseases in low and middle income countries. The inclusion of a wider geographical representation would improve the generalizability of the findings and capture diverse healthcare systems and economic burdens. Inconsistent Use of Economic Measurement Tools: The article mentions the lack of standardized economic evaluation methods across the reviewed studies. While this is a recognized issue, the authors could have elaborated further on how to overcome this inconsistency or suggested specific economic evaluation frameworks suitable for future research. Limited Discussion on Policy Implications: Although the study touches on the high economic burden faced by families, it would be beneficial to explore the policy implications in more depth. This could include recommendations for government assistance programs, insurance coverage reforms or advocacy efforts for better support for families dealing with rare diseases. Quality Assessment: The authors mention that a quality assessment of the included studies was not conducted due to the scoping review’s nature. However, a basic quality appraisal of key studies could still enhance the review’s rigor and provide insights into the reliability of the findings. Exclusion of Grey Literature: The decision to exclude grey literature, such as reports from patient organizations or government documents may limit the comprehensiveness of the review. Grey literature can provide valuable information, particularly regarding economic impacts and policy.

Use of AI tools: The general reading of the manuscript let me assured that the author has widely used chat GPT or other AI tools for writing this manuscript.

However, after careful consideration, I recommend rejecting this manuscript due to several critical issues. The article lacks depth in its analysis of the economic impacts and the methodology used for data extraction and synthesis is not sufficiently rigorous for publication. Additionally, the scoping review fails to provide any substantial new insights or advancements in the field instead reiterating previously known findings without offering a fresh perspective. The selection of studies appears biased toward Western countries which significantly limits the generalizability of the findings. Moreover, the lack of a quality appraisal of the included studies undermines the reliability of the conclusions drawn. Finally, the exclusion of grey literature results in an incomplete picture of the economic impacts and further weakens the overall contribution of paper to the field. For these reasons, I recommend that this manuscript be rejected in its current form.

Comments on the Quality of English Language

The author has widely used and relied on generative AI tools which makes this manuscript difficult to read at a number of places. I strongly recommend the author to write the research articles at their own or clearly explicit the use of AI tool as it is not within the ethical standards to use extensive AI tools in research.

Author Response

Thank you for your invaluable feedback on our manuscript. Your thorough review and the time taken to provide constructive comments are instrumental in shaping the final version of our work. We appreciate the opportunity to address your comments and provide clarification.

Comment 1: The article successfully covers a wide range of costs associated with pediatric rare diseases, contributing to the under-researched area of the economic burden on families.
Framework Application: The use of the PAGER framework is commendable as it provides a structured and systematic approach to reporting scoping reviews. This adds clarity to the results. Identification of Gaps: The paper highlights key research gaps in the field such as the lack of distinction between rare and ultra-rare diseases which is crucial for future research. The focus on the need for better economic evaluation tools is insightful.

Response 1: Thank you for highlighting these aspects of our manuscript.

Comment 2: Limited Geographical Representation: The studies predominantly focus on Western contexts (Europe and the United States) with very few addressing rare diseases in low and middle income countries. The inclusion of a wider geographical representation would improve the generalizability of the findings and capture diverse healthcare systems and economic burdens.

Response 2: Thank you for highlighting the limited scope of the study. The review had a global focus. The dominance of studies based in Western contexts reflects the studies that were returned by the search; however, this may have been influenced by the English language inclusion criterion. This limitation has been noted in the limitations section of this review. We have also noted that the presence of centres and networks such as European Reference Networks, which work to research rare diseases, may have contributed to the over-focus on western and English speaking contexts.

Comment 3: Inconsistent Use of Economic Measurement Tools: The article mentions the lack of standardized economic evaluation methods across the reviewed studies. While this is a recognized issue, the authors could have elaborated further on how to overcome this inconsistency or suggested specific economic evaluation frameworks suitable for future research.

Response 3: Thank you for your feedback. We would like to clarify that the objective of this manuscript is to provide an overview of the existing literature, rather than conduct an in-depth analysis of economic impacts or a detailed critique of methodologies used in the included studies. This article does not aim to advocate for one methodology over another; but rather to map current practice that can be expanded on in future research.

Comment 4: Limited Discussion on Policy Implications: Although the study touches on the high economic burden faced by families, it would be beneficial to explore the policy implications in more depth. This could include recommendations for government assistance programs, insurance coverage reforms or advocacy efforts for better support for families dealing with rare diseases.

Response 4: Thank you for your thoughtful feedback and for highlighting the importance of addressing policy implications in our study. We agree that exploring policy implications is critical in understanding and addressing the high economic burden families of children with rare diseases face.
While the primary focus of this scoping review was to map the existing literature, we recognize the value of providing more in-depth commentary on potential policy directions. In response to your suggestion, we will expand the discussion section to include a more detailed exploration of policy implications.

Comment 5: Quality Assessment: The authors mention that a quality assessment of the included studies was not conducted due to the scoping review’s nature. However, a basic quality appraisal of key studies could still enhance the review’s rigor and provide insights into the reliability of the findings.

Response 5: Thank you for your feedback regarding the lack of a quality assessment in the study. In response to these comments, an assessment of the quality of included articles using the MMAT assessment tool has been performed and included in the supplementary material accompanying this article.

Comment 6: Exclusion of Grey Literature: The decision to exclude grey literature, such as reports from patient organizations or government documents may limit the comprehensiveness of the review. Grey literature can provide valuable information, particularly regarding economic impacts and policy.

Response 6: Thank you for your feedback regarding the exclusion of grey literature. Given the scope of this review, we decided to focus on academic literature to highlight the state of research on the economic impacts of rare diseases within the academic context. Rare disease organizations and other grey literature sources are often specific to individual diseases within particular country contexts, which could have significantly broadened the scope of this review, making it unfeasible.

Comment 7: Use of AI tools: The general reading of the manuscript let me assured that the author has widely used chat GPT or other AI tools for writing this manuscript.

Response 7: Thank you for your comment on the language used throughout the manuscript. No AI tools, including Chat GPT, were used in the preparation of this manuscript, which was written by the primary author and reviewed by the remaining authors.

Comment 8: However, after careful consideration, I recommend rejecting this manuscript due to several critical issues. The article lacks depth in its analysis of the economic impacts and the methodology used for data extraction and synthesis is not sufficiently rigorous for publication. Additionally, the scoping review fails to provide any substantial new insights or advancements in the field instead reiterating previously known findings without offering a fresh perspective. The selection of studies appears biased toward Western countries which significantly limits the generalizability of the findings. Moreover, the lack of a quality appraisal of the included studies undermines the reliability of the conclusions drawn. Finally, the exclusion of grey literature results in an incomplete picture of the economic impacts and further weakens the overall contribution of paper to the field. For these reasons, I recommend that this manuscript be rejected in its current form.

Response 8: We thank you for your feedback. We would like to clarify that the objective of this manuscript is to present a scoping review. As is standard practice with scoping reviews, our primary aim is to map and provide an overview of the existing literature, rather than conduct an in-depth analysis of economic impacts or a detailed critique of methodologies used in the included studies. While we acknowledge that a rigorous examination of methods can provide valuable insights, this level of detail lies beyond the scope of this review. The methodology followed in this study adheres to established guidelines for conducting scoping reviews, ensuring a transparent and systematic approach to data extraction and synthesis.

The primary objective of this scoping review is to map the breadth of existing literature and synthesize the current state of knowledge on caregiver-reported economic impacts of pediatric rare diseases. By design, scoping reviews focus on consolidating available evidence and identifying research gaps rather than generating novel insights or proposing new advancements. This approach is particularly valuable in the context of rare diseases, where the evidence base is often scattered or fragmented due to the rarity and complexity of the conditions.
While the review does reiterate some known findings, this is a fundamental aspect of scoping reviews, as it ensures comprehensive coverage and clarity for readers. That said, we have made a concerted effort to highlight patterns, gaps, and underexplored areas within the literature. These insights can guide future research directions and inform policy or practice in this important area.

We hope this clarification addresses the concerns raised and highlights the intent and scope of the study. We are grateful for your constructive feedback, which will help us further refine our work to serve the field better.

Reviewer 2 Report

Comments and Suggestions for Authors

File attached

Author Response

Comment 1: Introduction: it would be beneficial to include a more in-depth discussion on the unique challenges faced by caregivers of children with rare diseases in managing both healthcare and non-healthcare costs, as this may differ significantly from adultfocused studies.

Response 1: Thank you for your insightful suggestion to expand the discussion on the unique challenges caregivers face of children with rare diseases, particularly in managing healthcare and non-healthcare costs.

In response to your feedback, we have incorporated additional literature discussing the distinct burdens caregivers face of children with rare diseases. This includes comparisons across different country contexts and highlights how parents of children with rare diseases often experience more significant challenges compared to caregivers of adults with rare diseases. These challenges encompass not only financial burdens but also emotional and logistical difficulties, which can differ significantly based on the age and dependency of the care recipient.

These additions will provide a more comprehensive introduction and establish the importance of this review.

Comment 2: Methods: the use of the CHEERS tool for quality appraisal may not have been optimal given the focus on cross-sectional studies rather than full economic evaluations

Response 2: Thank you for your considered feedback. Further detail on why the CHEERS tool was initially chosen for quality assessment is now included. 

Comment 3: The lack of an appropriate quality assessment tool could impact the reliability of findings

Response 3: Thank you for your feedback regarding a quality assessment of the included articles. In response to this comment, an assessment of the quality of included articles using the MMAT assessment tool has been performed and included in the supplementary material accompanying this article. 

Comment 4: it would be beneficial to clarify how discrepancies during the data charting process were addressed to ensure consistency

Response 4: Thank you for your feedback on the data charting process. In response to your suggestion, we have added a line to clarify how discrepancies during the data charting process were addressed. Specifically, we have noted that any discrepancies were resolved through discussion among the research team to ensure consistency and accuracy. This collaborative approach helped maintain the rigour and reliability of the data extraction process.

Comment 5: Could you provide further insights on the choice of rare disease-specific validated scales?

Response 5: Thank you for your thoughtful feedback. A line has been included on why the validated scales used were chosen by the authors. 

Comment 6: The results mention the economic impact of lost productivity and informal care but do not consistently quantify the financial implications.

Response 6: Thank you for your comment highlighting the lack of consistent quantification of productivity loss and informal care. This lack of consistency in quantification of financial implications of loss of productivity and informal care is a finding of this review.

Comment 7: Without validated, universally accepted tools, it is challenging to accurately assess and compare the financial burden across different studies and populations. This gap limits the ability of this manuscript. Explain

Response 7: Thank you for highlighting the challenge posed by the lack of validated and universally accepted tools for accurately assessing and comparing financial burdens across studies and populations.

While we agree that such comparisons are currently not feasible due to the absence of standardized tools, it is important to note that this review does not aim to perform such comparisons. Instead, this review aims to identify and highlight the types of costs experienced by families of children with rare diseases. By mapping these costs, we hope to contribute foundational knowledge that can inform the development of validated tools, which may enable such comparisons in the future.

Reviewer 3 Report

Comments and Suggestions for Authors

I read with interest the article by Buckle et al. on the economic impacts of pediatric rare diseases.  It is a well-designed and well-conducted study. However, a couple of similar studies exist. I have a few comments to improve clarity.

Please address the discrepancy regarding the search date in the abstract and methods section. Although a year has passed since the search date, an updated search could potentially enhance the results; however, given the nature of this study, the authors can decide whether or not to pursue it.

I would also recommend adding subsections within the methods section to improve readability.

Finally, I suggest discussing the association between social determinants of health and the economic impact of rare diseases.

Author Response

Comment 1: Please address the discrepancy regarding the search date in the abstract and methods section. Although a year has passed since the search date, an updated search could potentially enhance the results; however, given the nature of this study, the authors can decide whether or not to pursue it.

Response 1: Thank you for pointing out this error. This has been corrected in the abstract to reflect the correct date. 

Comment 2: I would also recommend adding subsections within the methods section to improve readability.

Response 2: Thank you for your recommendation to add subsections within the methods section. The following sub-headings have been added in response to this comment – ‘Inclusion and Exclusion Criteria’, ‘Procedure’, ‘Quality Assessment’.

Comment 3: Finally, I suggest discussing the association between social determinants of health and the economic impact of rare diseases.

Response 3: Thank you for suggesting that we discuss the association between social determinants of health and the economic impact of rare diseases. This is indeed a critical area of consideration, as factors such as income, education, geographic location, and access to healthcare can significantly influence the financial burden experienced by families.

In response to your feedback, we will expand the discussion section to include insights into how social determinants of health intersect with the economic impacts of rare diseases. We believe this addition will enrich the manuscript and provide a broader context for understanding the economic challenges faced by caregivers.

Reviewer 4 Report

Comments and Suggestions for Authors

Dear editor,

The paper addresses a highly interesting topic and is well-structured and thoroughly developed. The methodological section is particularly clear and comprehensive. However, the descriptive section would benefit from additional graphical or tabular descriptive analyses to improve its presentation. Overall, the paper is well-written, and it was a pleasure to read. I recommend the publication with minor revisions.

Minor Comments:

1.     The author should more clearly specify in the text the exact search string used, including details on the field selection. The large initial sample size appears to be due to insufficiently defined fields of interest.

2.     I also wonder why the author did not employ a snowballing search approach to identify additional relevant articles.

3.     Section 3.1 would benefit from greater use of graphical or tabular descriptive analyses. Specifically, Section 3.1.3 is challenging to follow and could be improved with a summary table highlighting the main elements. It would also be beneficial to indicate the sample analyzed in each article.

4.      A table summarizing the typologies of costs observed (e.g., direct and indirect costs, whether related to children or their carers) would be also a value added.

5.     Lastly, there seems to be a typo in Table 1, “Article Summary Table.” Should the article number listed as 19 be corrected to 18?

Author Response

Comment 1: The author should more clearly specify in the text the exact search string used, including details on the field selection. The large initial sample size appears to be due to insufficiently defined fields of interest.

Response 1: Thank you for your feedback. The search strategy for each database search is now provided in tabular format in the supplementary material.

Comment 2: I also wonder why the author did not employ a snowballing search approach to identify additional relevant articles.

Response 2: Thank you for your interest in the search approach used. This review did not employ a snowballing approach due to the high volume of articles returned in the database searches, which exceeded 20,000. The comprehensive search string effectively identified the most relevant articles, and we did not encounter significant gaps or "hidden" studies that would have warranted additional snowballing.

Additionally, we were mindful of the potential for increased risk of bias associated with snowballing, particularly in cases where interconnected studies may dominate the dataset. Many of the studies included in the review were already linked or cross-referenced, reducing the necessity of employing this approach.

We hope this explanation provides clarity regarding this methodological decision.

Comment 3: Section 3.1 would benefit from greater use of graphical or tabular descriptive analyses. Specifically, Section 3.1.3 is challenging to follow and could be improved with a summary table highlighting the main elements. It would also be beneficial to indicate the sample analyzed in each article.

Response 3: Thank you for your advice regarding section 3.1. This section has been adapted to a tabular format and graphics for the year of publication and country contexts have been inserted. The sample characteristics for each article are included in the newly inserted Table 2.

Comment 4: A table summarizing the typologies of costs observed (e.g., direct and indirect costs, whether related to children or their carers) would be also a value added.

Response 4: Thank you for your feedback regarding the typologies of costs observed. Table 1. includes the economic impact categories and specific impacts measured by each of the articles. Financial costs are related to the child’s rare disease experienced by parents. The costs have been divided by those relating to the child and those relating to the caregiver.

Comment 5: Lastly, there seems to be a typo in Table 1, “Article Summary Table.” Should the article number listed as 19 be corrected to 18?

Response 5: Thank you for your feedback. Apologies, but I cannot see which article is listed as 19 in text – there is only one listed as 18. This article number is listed correctly as it is referenced in-text.

Reviewer 5 Report

Comments and Suggestions for Authors

Dear Authors,

I really appreciate your paper. In order to improve it, please find below my comments:

1. You used only one definition for rare diseases. I suggest to specify that there are different level. FRare, that  means one out of 5000-10000, very rare , that means one out of 10000-50000, hyper rare, one out of over 50.000. These different cases are relevant or not relevant for your analysis? Please clarify.

2. It will be helpful to underline if there are differences in the composition of direct costs, indirect costs for families, productivity losses, for different typologies of rare diseases. It will be helpful to underline in the discussion paragraph.

3. I wonder if you noticed different patient approaches in different countries for the same disease. I refer to the comparative studies you analyzed.

4. Do you find different mix for the same rare disease in the same country? If so, can you try to interpret the differences?

5. How do you define cathastrophic expenditure? which definition you refer to? Is it a percentage for family income.

6. I also suggest to specify the significance of different paper you analyzed. I mean the number of families involved because there is a difference between a survey for 10-20 families and a survey for 50 or 100 families.

7. I suggest to say something about the different logic or different policies towards rare diseases you perceived. I mean some countries in which there are few specialized centers (hub), others in which there is a network of center for each rare disease. 

Author Response

Comment 1: You used only one definition for rare diseases. I suggest to specify that there are different level. FRare, that  means one out of 5000-10000, very rare , that means one out of 10000-50000, hyper rare, one out of over 50.000. These different cases are relevant or not relevant for your analysis? Please clarify.

Response 1: Thank you for your considered suggestion to include further information on rare disease classifications. More discussion on rare/ ultra-rare/ hyper rare diseases and why these classifications were not utilised in the review has been added to the introduction in response.

Comment 2: It will be helpful to underline if there are differences in the composition of direct costs, indirect costs for families, productivity losses, for different typologies of rare diseases. It will be helpful to underline in the discussion paragraph.

Response 2: Thank you for your feedback. Table 1. includes the economic impact categories and specific impacts measured for families of children of different rare diseases in each of the articles. Financial costs are related to the child’s rare disease experienced by parents. The costs have been divided by those relating to the child and those relating to the caregiver. It is important to highlight that this article only aims to map the costs measured in each article, rather than determine which costs are experienced by different rare diseases.

Comment 3: I wonder if you noticed different patient approaches in different countries for the same disease. I refer to the comparative studies you analyzed.

Response 3: Thank you for your comment. Can you please clarify by what is meant by ‘patient approaches’, whether it is related to research or health services?

Comment 4: Do you find different mix for the same rare disease in the same country? If so, can you try to interpret the differences?

Response 4: Thank you for your thoughtful feedback. Many of the studies did not overlap in terms of rare disease and country, except for two studies that crossed over (Flores et al., 2020 and Rodriguez et al., 2022, Spain, Duchenne Muscular Dystrophy). However, these studies looked at different elements of economic impacts. Flores et al. (2020) focused on quantification of economic impact and predictors of increased economic caregiver burden; Rodríguez et al. (2022) looked at caregiver burden related to wellbeing and compared a Spanish cohort to a similar cohort in Mexico. Therefore, a comparison of the two studies was not included in the review.

Comment 5: How do you define cathastrophic expenditure? which definition you refer to? Is it a percentage for family income.

Response 5: Thank you for your comment. The definition of catastrophic health expenditure, as defined by the relevant study, is now included.

Comment 6:  I also suggest to specify the significance of different paper you analyzed. I mean the number of families involved because there is a difference between a survey for 10-20 families and a survey for 50 or 100 families.

Response 6: Thank you for your thoughtful feedback. The sample characteristics for each article is now included in tabular format. Regarding significance, as there is a mix of study designs (cross-sectional, qualitative, etc.) the data generated across each study varies in its depth and scope, therefore I am not sure how to specify the significance of each article.

Comment 7: I suggest to say something about the different logic or different policies towards rare diseases you perceived. I mean some countries in which there are few specialized centers (hub), others in which there is a network of center for each rare disease.

Response 7: Thank you for your feedback. Further discussion on the policy implications of the economic impacts of rare diseases on families has been included in the review. The introduction of rare disease centres and the work of projects like JARDIN that can improve rare disease care and HCP’s knowledge of the economic impacts of these conditions have been acknowledged.

Round 2

Reviewer 1 Report

Comments and Suggestions for Authors

The current revised version of the manuscript has improved the quality of the manuscript. I recommend to accept the manuscript in its current form for publication in healthcare journal. Best wishes to the authors.

Comments on the Quality of English Language

In revised version, the quality of communication is improved.

Reviewer 2 Report

Comments and Suggestions for Authors

well revised